# Dream It, Do It? Associations between Pornography Use, Risky Sexual Behaviour, Sexual Preoccupation and Sexting Behaviours among Young Australian Adults

**Elizabeth Mary Clancy** [1,2,*] **, Dominika Howard** [1,2] **, Shaoyuan Chong** [3,4] **and Bianca Klettke** [1,2,*]

1   School of Psychology, Faculty of Health, Deakin University, Geelong 3220, Australia;
    Dominika.howard@deakin.edu.au
2   Centre for Social and Early Emotional Development, Deakin University, Geelong 3220, Australia
3   Department of Psychology, Faculty of Arts and Social Sciences, University of British Columbia (Okanagan),
    Kelowna, BC V1V 1V7, Canada; chongshaoyuan@gmail.com
4   Department of History and Sociology, Faculty of Arts and Social Sciences,
    University of British Columbia (Okanagan), Kelowna, BC V1V 1V7, Canada
*   Correspondence: elizabeth.clancy@deakin.edu.au (E.M.C.); bianca.klettke@deakin.edu.au (B.K.);
    Tel.: +61-392-446-207 (B.K.)

**Abstract:** While sexting behaviours have attracted increasing research focus over the last decade as both normative and deviant forms of sexual activity, little attention has been paid to their potential associations with sexual preoccupation and heightened interest in sex. The current study sought to identify whether sexual preoccupation significantly predicts sending, receiving, and disseminating sexts, after controlling for pornography use and risky sexual behaviours. Young Australian adult participants (*N* = 654, 78.8% women) aged 18 to 34 (*M* = 19.78, SD = 1.66) completed an anonymous online self-report questionnaire regarding their engagement in sexting behaviours (sending, receiving, and dissemination), pornography use, risky sexual behaviours, and sexual preoccupation. Results showed that individuals with higher sexual preoccupation were more likely to engage in pornography use and risky sexual behaviours. Binary hierarchical logistic regressions revealed that sexual preoccupation predicted higher rates of sending and receiving sexts. However, sexual preoccupation did not significantly contribute to increased rates of sext dissemination. Our study illustrates the need to incorporate pornography viewing and sexting into the promotion of safe sexual behaviours in online and offline contexts, and the potential to utilise modern technology to negotiate safer sex practices.

**Keywords:** sexting; sexual preoccupation; pornography; sexual behaviour; emerging adulthood

## 1. Introduction

With the growth of digital technology, people have changed the way they relate to and communicate with others [1,2]. In particular, this includes engagement in sexting behaviours [3,4] defined as the sending, receiving, or forwarding of sexually explicit messages, images, or videos via electronic means [4]. This digitally mediated sexual communication is popular among young adults, with the mean prevalence rates for sending estimated at approximately 38%, for receiving at 41%, and for two-way sexting (sending and receiving) at 48% [5]. Additionally, meta-analytic findings suggest that men tend to request and women to send sexts more frequently, but there is no consistent gender difference in two-way sexting [5]. Whilst a number of factors have been explored to explain engagement in sexting behaviours [4], little is known about how individuals with increased sexual preoccupation may use sexting to channel their sexual needs and desires.

## 1.1. A Brief Sexting Overview

To date, research on sexting has predominantly focused on the potential social, psychological, and legal consequences of this behaviour [4,6,7]. On the social level, individuals who sext are often perceived as "morally corrupt" and consciously risking their good reputation [8]. Some individuals report feeling pressured/coerced to sext, or having had their private images shown or forwarded to others without their knowledge or consent [9–13]. These coercive and non-consensual sexting experiences have been associated with worse mental health functioning and symptoms of stress, anxiety, and depression [6,10,14], and hence are recognised as harmful. From a legal perspective, some sexting behaviours may lead to arrest and conviction. For example, forwarding/dissemination of sexts and even threats to disseminate such material are criminalized under English, American, and Australian legislation [6,7].

Whilst sexting, particularly among adolescent populations or when coerced/unwanted, has received significant negative media coverage and research attention, some scholars argue that consensual and wanted sexting can be healthy and normative [8,9]. Sexting behaviours are often underpinned by the desire to flirt and be fun, attract a partner, "hook up", and express one's sexual needs and fantasies—e.g., [8,10–12]. Unsurprisingly then, sexting is frequently associated with offline sexual activities, with sexters reporting a greater number of sexual partners when compared to those who do not engage in sexting [13–15]. Additionally, some research has found sexting to be associated with risky sexual behaviours encompassing inconsistent condom use, unprotected sex, and substance use during sexual activity [3,16–18]. Other studies have found associations among sexting, alcohol consumption, substance use, and risk-taking more broadly [19–22], particularly in adolescence. Research by Morelli and colleagues [23] revealed that alcohol consumption moderated the relationship between cyberpornography and sexting, with higher consumption of alcohol being associated with stronger associations between cyberpornography and sending sexts. However, these findings are not always consistent, with some studies reporting that receiving sexts was associated with both unprotected sex and condom use [24], and other findings being mixed [25].

In addition to offline sexual activities, sexting has been associated with a number of other online sexual activities including cybersex, chatting online with strangers, using sex sites, and viewing online pornography [3,23,26–29]. Pornography in particular is relatively ubiquitous among youths and emerging adults. Australian research 15 years ago [30] suggested that among older adolescents, 75% of boys and 10% of girls had been exposed to X-rated movies, whilst three quarters had accidentally been exposed to pornographic websites. More recent research [31] from population-representative adult studies reports increased rates of online pornography consumption: for those aged 16–19 years, 93.4% of men and 72.5% of women had used pornography in the past year, while among 20–29 year-olds, 88.6% of men and 66.3% of women viewed pornographic material. These online behaviours often serve to facilitate and enhance sexual experiences and fulfil sexual desires, especially for those who exhibit a propensity to be overly preoccupied with sex, or engage in risky sexual behaviours [4,17,26,32–35]. Sexting, online sexual behaviours, and pornography consumption have often been conceptualised as interchangeable by some researchers, with these activities being perceived as a safe means to relieve sexual tension [29].

## 1.2. Sexual Preoccupation

One predisposing factor for engagement in online sexual activities is sexual preoccupation [36,37]. Sexual preoccupation refers to an abnormally intense interest in sex and a form of sexual deviancy associated with an increased interest in and frequency of sexual fantasies, thoughts, and activities [38–41]. Individuals who exhibit sexual preoccupation are more likely to engage in masturbation and sexual intercourse, as well as risky sexual behaviours (e.g., sex with one-time partners, condomless vaginal or anal sex) in an effort to resolve distress and anxieties extending from excessive sexual thoughts [40,42]. Individuals

reporting higher rates of sexual preoccupation have been found to engage in more frequent online sexual activities such as pornography consumption [38,43,44]. Concerningly, sexual preoccupation has also been consistently associated with recidivist violence and sexual offending [39–41,45,46].

Despite the close associations found between sexual preoccupation and online sexual activities, engagement in sexting behaviours among individuals with higher levels of sexual preoccupation has received minimal research attention. To date, only three studies have investigated sexual preoccupation or compulsion and sexting behaviours, with contradictory findings. Howard, Klettke, Clancy, Fuelscher, and Fuller-Tyszkiewicz [32] found that sexual preoccupation predicted a greater willingness to send sexually explicit nudes or seminudes. In contrast, Perkins, Becker, Tehee, and Mackelprang [14] found no relationship between sending nude or seminude photographs, pornography use, and sexual compulsion, a phenomenon similar to sexual preoccupation whereby an individual finds it difficult to control his/her drive for sexual activity [14]. Trendell [47] revealed that sexual compulsivity was positively associated with harmful sexting behaviours, specifically perpetration of sexting coercion and the non-consensual distribution of intimate images.

It is noteworthy that all three studies used different measures of sexual preoccupation and investigated various aspects of sexting behaviours, rendering direct comparison of the results difficult. Specifically, whilst both Howard, Klettke, Clancy, Fuelscher, and Fuller-Tyszkiewicz [32] and Perkins, Becker, Tehee, and Mackelprang [14] investigated the sending of sexts, Trendell [47] examined coerced sexting and non-consensual sext dissemination. Nonetheless, the results suggest that sexual preoccupation may be implicated in explaining some sexting behaviours, including those that may be pernicious in nature (sext dissemination). As such, more research is required to replicate and extend current knowledge on these associations.

*1.3. Gaps in the Literature*

Despite growing research pertaining to motivations and risk factors associated with sexting behaviours, little is known about the associations between sexual preoccupation and specific sexting behaviours. In particular, there are established bivariate relationships among sexting and risky offline sexual behaviours, online sexual activities, and sexual preoccupation. However, it is unclear as to whether these relationships overlap or whether sexual preoccupation provides an additional explanation of sexting behaviours, over and above those established with offline and online behaviours. In addition, there are mixed findings as to whether individuals rated high on sexual preoccupation are more likely to send sexts and to disseminate sexts, with no research exploring the association between sexual preoccupation and receiving sexts. Therefore, the investigation of sending, receiving, and dissemination would be of value. Thirdly, existing studies investigating associations between pornography use and sexting assess only limited forms of pornography and rarely use validated scales to assess pornography consumption. As previously discussed, sexual preoccupation has been found to be associated with forensic outcomes, including being involved in recidivist violent and sexual offending behaviour. Given that sexting behaviours can be associated with online sexual abuse, e.g., through non-consensual sext dissemination, there is a need for clearer indications of whether sexual preoccupation as a trait is indeed associated with this form of sexting behaviour. Such knowledge may offer opportunities for focused treatment and prevention measures to mitigate the risks.

*1.4. Aims and Hypotheses*

Extending on previous literature arguing for the close association among sexual preoccupation and online and offline sexual activities, this study seeks to explore the associations among sexting and sexual preoccupation, pornography use, and risky sexual behaviours. Based on prior findings that those reporting higher rates of sexual preoccupation engage in more frequent online sexual activities [38,43,44], we anticipate that individuals with higher traits of sexual preoccupation will be (1) more likely to engage in

pornography use and offline risky sexual behaviours. After controlling for pornography use and risky sexual behaviours, which have been found to be associated with sexting behaviours [3,4,17,23,26–29,33], and based on the limited available research investigating these relationships [14,32,47], we expect that individuals with higher levels of sexual preoccupation will be (2) more likely to send sexts, (3) more likely to receive sexts, and (4) more likely to forward/disseminate sexts.

**Hypothesis 1.** *That individuals with higher traits of sexual preoccupation will be more likely to engage in pornography use and offline risky sexual behaviours.*

**Hypothesis 2.** *That individuals with higher traits of sexual preoccupation will be more likely to send sexts.*

**Hypothesis 3.** *That individuals with higher traits of sexual preoccupation will be more likely to receive sexts.*

**Hypothesis 4.** *That individuals with higher traits of sexual preoccupation will be more likely to forward/disseminate sexts.*

## 2. Materials and Methods

### 2.1. Participants

A convenience sample of 654 Australian participants aged 18 to 34 was recruited for this study ($M = 19.78$, $SD = 1.66$). Of this sample, 21.3% ($n = 139$) identified as men and 78.7% ($n = 515$) as women. Regarding sexual orientation, 74.6% identified as heterosexual, 18.5% as bisexual, 5.7% as homosexual, and 1.2% declined to state their sexual identity. Of the participants, 80.7% reported being sexually active ($n = 528$), with a mean age of first sexual intercourse of 16.54 years ($SD = 1.75$).

### 2.2. Measures

#### 2.2.1. Sexting Behaviours

Sexting behaviours were assessed using survey items drawn from Clancy et al. [48], specifically participant self-reports of whether they had ever "sent" or "received sexually explicit images via text message or a mobile app". Responses were recorded dichotomously as "yes" or "no". To assess coercion, participants were asked if they had "ever asked others to send them sexually explicit images of themselves", with responses recorded as "yes" or "no". Lastly, participants were asked if they had "ever shared sexts with others even though the sexts were intended for themselves", with responses recorded dichotomously as "yes" or "no".

#### 2.2.2. Risky Sexual Behaviours

In order to assess if participants engaged in risky sexual behaviours, the Sex-Risk subscale of the Adolescent Risk Inventory [49] was used. This seven-item questionnaire includes two interval and five dichotomous items. Interval questions related to the frequency of sexual intercourse in the past year (response options: "None", "1–5 times", or "6 or more times"), and the number of different sex partners in the past year (response options: "None", "one", or "two or more"). Dichotomous items addressed specific sexual risk behaviours with "yes" vs. "no" responses permitted, including whether participants ever had sex without a condom; if they used alcohol or drugs during sex; and if they ever had a sexually transmitted disease. A sex risk score was established by totalling all seven items. Internal reliability in this study was good, with a Cronbach's alpha of 0.76. This is comparable with the initial validation study [49], which reported a Cronbach's alpha of 0.72 in a population of U.S. adolescents (aged 12–19) with psychiatric disorders. Whilst the scale was initially validated in adolescents, the set of questions was deemed to provide a good measure of sexual risk amongst young adult population.

### 2.2.3. Pornography Consumption Questionnaire

To evaluate rates of adult pornography consumption, two questions were drawn from the Pornography Consumption Questionnaire [50]. This self-report measure consists of five questions concerning pornography consumption frequency, recency, and intensity, where pornography is defined as "any kind of sexually explicit material displaying genitalia with the aim of sexual arousal or fantasy". An initial dichotomous question asked participants if they have "ever watched adult pornography" ("yes" vs. "no"). If yes, they were then asked about the frequency of their use ("once a month or less", "1–2 days per month", "1–2 days a week", "3 to 4 days a week", and "every day or almost every day"), as well as recency and intensity of use, as well as the age of first exposure. The PSQ has demonstrated reliability, with a Cronbach's alpha of 0.87 in prior versions [51]. In this study, questions were analysed individually to explore lifetime usage and current frequency of use.

### 2.2.4. Sexual Preoccupation Subscale

For the purposes of measuring sexual preoccupation, this study employed a 10-item Sexual Preoccupation Subscale of the Sexuality Scale [52]. Participants indicated their agreement with the items, e.g., "I think about sex more than anything else", on a centred 5-point Likert scale ranging from −2 (disagree) to 2 (agree). After reverse-coding, the scores were added with higher aggregated scores reflective of a higher level of sexual preoccupation. Prior reported internal consistency for the scale was good [53]. Results in this study demonstrated excellent internal consistency, with Cronbach's alpha of 0.91.

### 2.3. Procedure

Following approval from the Deakin University Human Research Ethics Committee (Reference HEAG-H 96-2012), participants were recruited via personal social media pages, Facebook advertisements, email distribution, and university campus advertisements over an eight-week period. Participants were also encouraged to share the study further via snowball recruitment. Advertisements advised participants that the survey discussed topics relating to their sexting, online and offline sexual activities. After clicking on the Qualtrics survey link, participants were provided with a plain language statement outlining the purpose of the study, confidentiality of responses, and support options. After providing informed consent, participants provided demographic information. Based on an a priori power analysis [54], with a small-to-medium effect size, a minimum sample size of $N = 393$ was deemed necessary to achieve 80% power. In total, 656 participants filled out the online survey over a two-month period. Of these, two responses were removed as they exceeded the inclusion criteria (age 18–35 years). The online survey took approximately 20–25 min to complete. Participation was voluntary and anonymous, and no financial incentives were provided.

### 2.4. Data Analysis

After completion of data collection, data were analysed using IBM SPSS V25 (IBM: Armonk, NY, USA). Bivariate correlations were measured to establish associations among individual variables (sending, receiving, and disseminating sexts, sexual preoccupation, pornography viewing, and risky sexual behaviour). Thereafter, binary hierarchical logistic regressions were conducted to establish if higher rates of sexual preoccupation, pornography viewing, and risky sexual behaviour predicted increased engagement in sext sending, receiving, and dissemination.

## 3. Results

Table 1 shows descriptive statistics for the study variables, stratified by gender and dichotomized sexual orientation. Most of the sample had both received and sent sexts, with no gender difference. However, those identifying as sexual minorities were more likely to engage in sending and receiving sexts than heterosexual participants. Almost one-quarter of respondents had disseminated sexts to others, with men more likely to disseminate

sexts than women and no difference by sexual orientation. The majority of the sample was sexually active in the last year, and a third had multiple sexual partners in this timeframe. When examining risky sexual behaviours, more than half the sample had engaged in sex without a condom, with women more likely than men to do so, and around half reported having had sex while under the influence of alcohol or drugs. However, when examined overall, there was no significant difference in the rates of risky sexual behaviours by either gender or sexual orientation. Three-quarters of participants had watched pornography (men more than women and non-heterosexual participants more than those identifying as heterosexual). Rates of use were skewed to less frequent engagement, particularly for women and heterosexual participants. Sexual preoccupation rates were higher for men than women and for non-heterosexual participants compared to heterosexual participants. These rates were within the intermediate range for men and the high range for women, based on published norms [52], with no norms available for sexual orientation.

**Table 1.** Descriptive statistics for key variables of interest.

| Variable | Full Sample (N = 654) | Men (N = 140) | Women (N = 516) | $\chi^2$ | $\varphi$ | Hetero (N = 488) | Non-Hetero (N = 166) | $\chi^2$ | $\varphi$ |
|---|---|---|---|---|---|---|---|---|---|
| | % | % | % | | | | | | |
| Have received sexts | 83.0 | 81.3 | 83.5 | 0.37 | 0.02 | 81.1 | 88.6 | 4.86 * | 0.09 |
| Have sent sexts | 72.3 | 66.9 | 73.9 | 2.70 | 0.06 | 70.0 | 79.5 | 5.59 * | 0.09 |
| Have disseminated sexts | 23.4 | 34.5 | 20.4 | 12.22 *** | −0.14 | 23.6 | 22.9 | 0.03 | −0.01 |
| Watch pornography | 75.7 | 95.7 | 70.3 | 38.35 *** | 0.24 | 72.1 | 86.1 | 13.22 *** | 0.14 |
| Frequency of watching pornography * | | | | | | | | | |
| Once a month or less | 37.3 | 7.4 | 48.4 | 142.94 *** | | 40.8 | 28.5 | 12.35 * | 0.16 |
| 2–3 d a month | 17.7 | 7.2 | 21.6 | | | 15.4 | 23.6 | | |
| 1–2 d a week | 26.1 | 39.0 | 21.3 | | | 26.8 | 24.3 | | |
| 3–5 d a week | 9.4 | 21.3 | 4.9 | | | 9.2 | 9.7 | | |
| Everyday/almost everyday | 9.6 | 25.0 | 3.8 | | | 7.8 | 13.9 | | |
| | M (SD) | M (SD) | M (SD) | t | p | M (SD) | M (SD) | t | p |
| Risky sexual behaviours | 4.19 (2.35) | 3.91 (2.47) | 4.27 (2.32) | −1.61 | 0.11 | 4.09 (2.39) | 4.50 (2.21) | −1.96 | 0.05 |
| Sexual preoccupation | 4.51 (9.04) | 6.63 (8.23) | 3.93 (9.16) | 3.15 | 0.00 | 4.08 (8.99) | 5.75 (9.07) | −2.03 | 0.04 |

Note. Counts and percentages refer to participants who answered in the affirmative. * $p < 0.05$; *** $p < 0.001$. Hetero = participants selecting heterosexual as their sexual orientation, Non-hetero = participants selecting homosexual, bisexual, or preferring not to state their orientation.

Table 2 shows bivariate correlations among the study variables. Most online and offline sexual behaviours were significantly and positively correlated with receiving, sending, and disseminating sexts.

**Table 2.** Bivariate correlations among study variables.

| Variable | N | 1 | 2 | 3 | 4 | 5 | 6 | 7 | 8 | 9 | 10 |
|---|---|---|---|---|---|---|---|---|---|---|---|
| 1. Age | 654 | | | | | | | | | | |
| 2. Gender | 654 | −0.07 | | | | | | | | | |
| 3. Sexual orientation | 654 | 0.08 * | 0.02 | | | | | | | | |
| 4. Sent a sext | 653 | 0.07 | 0.06 | 0.09 * | | | | | | | |
| 5. Received sext | 653 | 0.05 | 0.02 | 0.09 * | 0.59 ** | | | | | | |
| 6. Disseminated sext | 654 | 0.04 | −0.14 ** | −0.01 | 0.23 ** | 0.16 ** | | | | | |
| 7. Sexually active | 654 | 0.14 | 0.10 * | 0.09 * | 0.38 ** | 0.34 ** | 0.11 ** | | | | |
| 8. View adult pornography | 654 | 0.01 | −0.24 ** | 0.14 ** | 0.21 ** | 0.19 ** | 0.13 ** | 0.15 * | | | |
| 9. Frequency of pornography use | 502 | −0.03 | −0.52 ** | 0.10 * | 0.03 | 0.04 | 0.15 ** | −0.08 | 0.08 | | |
| 10. Risky sexual behaviours | 654 | 0.20 ** | 0.06 | 0.08 | 0.44 ** | 0.36 ** | 0.00 | 0.23 ** | 0.72 ** | 0.14 ** | |
| 11. Sexual Preoccupation | 654 | −0.02 | −0.12 ** | 0.08 * | 0.29 ** | 0.23 ** | 0.14 ** | 0.20 ** | 0.28 ** | 0.29 ** | 0.27 ** |

Note. * $p < 0.05$, ** $p < 0.01$. Significance of Pearson correlations based on two-tailed significance.

In line with Hypothesis 1, individuals with higher sexual preoccupation were more likely to engage in pornography use, risky sexual behaviours, and sexting behaviours. However, the strength of these correlations was small [55].

*3.1. Main Analyses*

3.1.1. Sending Sexts

A binary hierarchical logistic regression was used to test whether individuals with higher traits of sexual preoccupation were more likely to send a sext (Hypothesis 2), with the results presented in Table 3. Entering age, gender, and sexual orientation in Step 1, the model significantly predicted sending sexts: $\chi^2(3) = 13.32$, $p = 0.004$. Inclusion of pornography viewing and risky sexual behaviours in Step 2 significantly improved the model, with the variables significantly predicting sending sexts: $\chi^2(5) = 150.19$, $p < 0.001$. In Step 3, the addition of sexual preoccupation further improved the model fit, $\chi^2(6) = 173.11$, $p < 0.001$, supporting Hypothesis 2. In addition to sexual preoccupation, gender, pornography viewing, and risky sexual behaviours constituted unique significant predictors in the overall model.

**Table 3.** Results of logistic regressions for sexting behaviours (yes/no).

| Variable | | | | | 95% CI | | | |
|---|---|---|---|---|---|---|---|---|
| | *B* | *df* | *p* | Exp(*B*) | Lower | Higher | $R^2_{LL}$ | $R^2_{Change}$ |
| **Sent Sext** | | | | | | | | |
| Step 1 | | | | | | | 0.01 * | |
| Step 2 | | | | | | | 0.19 *** | 0.19 *** |
| Step 3 | | | | | | | 0.22 *** | 0.03 *** |
| Constant | −2.942 | 1 | 0.042 | 0.53 | − | − | | |
| Age | 0.02 | 1 | 0.730 | 1.02 | 0.90 | 1.17 | | |
| Gender | 0.68 | 1 | 0.008 | 1.97 | 1.20 | 3.25 | | |
| Pornography viewing | 0.79 | 1 | 0.001 | 2.21 | 1.39 | 3.53 | | |
| Risky sexual behaviour | 0.40 | 1 | <0.001 | 1.49 | 1.36 | 1.63 | | |
| Sexual preoccupation | 0.06 | 1 | <0.001 | 1.06 | 1.03 | 1.09 | | |
| **Received Sext** | | | | | | | | |
| Step 1 | | | | | | | 0.00 | |
| Step 2 | | | | | | | 0.13 *** | 0.13 *** |
| Step 3 | | | | | | | 0.14 *** | 0.01 ** |
| Constant | −0.90 | 1 | 0.59 | 0.41 | | | | |
| Age | 0.00 | 1 | 0.94 | 0.99 | 0.86 | 1.16 | | |
| Gender | 0.40 | 1 | 0.18 | 1.49 | 0.84 | 2.64 | | |
| Pornography viewing | 0.74 | 1 | 0.005 | 2.10 | 1.26 | 3.52 | | |
| Risky sexual behaviour | 0.36 | 1 | <0.001 | 1.43 | 1.30 | 1.58 | | |
| Sexual preoccupation | 0.05 | 1 | 0.001 | 1.05 | 1.02 | 1.08 | | |
| **Disseminated Sext** | | | | | | | | |
| Step 1 | | | | | | | 0.02 * | |
| Step 2 | | | | | | | 0.06 *** | 0.05 *** |
| Step 3 | | | | | | | 0.08 *** | 0.03 |
| Constant | −1.00 | 1 | 0.43 | 0.37 | | | | |
| Age | −0.03 | 1 | 0.662 | 0.98 | 0.87 | 1.09 | | |
| Gender | −0.74 | 1 | 0.001 | 0.48 | 0.30 | 0.75 | | |
| Pornography viewing | 0.40 | 1 | 0.139 | 1.49 | 0.88 | 2.54 | | |
| Risky sexual behaviour | 0.27 | 1 | <0.001 | 1.30 | 1.18 | 1.44 | | |
| Sexual preoccupation | 0.02 | 1 | 0.155 | 1.02 | 0.99 | 1.04 | | |

Note. *N* = 654. All variables coded such that endorsement of yes or higher scores indicates a higher risk of the relevant behaviour. * $p < 0.05$, ** $p < 0.01$, *** $p < 0.001$.

3.1.2. Receiving Sexts

To test whether individuals with higher traits of sexual preoccupation were more likely to receive sexts (Hypothesis 3), we ran a second binary hierarchical regression analysis. Inclusion of age, gender, and sexual orientation in Step 1 was significant: $\chi^2(3) = 8.05.44$, $p = 0.045$. Inclusion of pornography viewing and risky sexual behaviours in Step 2 significantly improved the model fit for predicting receiving sexts: $\chi^2(5) = 98.53$, $p = < 0.001$. Inclusion of sexual preoccupation in Step 3 further improved the model fit; therefore, Hypothesis 3 was supported. The set of predictors significantly predicted receiv-

ing sexts, $\chi^2(6) = 109.88$, $p < 0.001$, with sexual preoccupation, pornography viewing, and risky sexual behaviours all constituting unique significant predictors for the overall model.

### 3.1.3. Sext Dissemination

To test whether individuals with higher traits of sexual preoccupation were more likely to forward (disseminate) sexts (Hypothesis 4), we ran a third binary hierarchical regression analysis. In Step 1, entering age, gender, and sexual orientation resulted in a significant model: $\chi^2(3) = 11.92$, $p = 0.008$. Inclusion of pornography viewing and risky sexual behaviours significantly improved the model, and this set of variables significantly predicted sext dissemination: $\chi^2(5) = 56.56$, $p < 0.001$. Inclusion of sexual preoccupation in Step 3 did not significantly improve the model; therefore, Hypothesis 4 was not supported. However, the set of variables significantly predicted disseminating sexts, $\chi^2(5) = 57.72$, $p < 0.001$, with gender and risky sexual behaviours being unique significant predictors of sext dissemination.

## 4. Discussion

The objective of the current study was to investigate the degree to which risky sexual behaviours, sexual risk-taking, and sexual preoccupation explain various forms of sexting: sending, receiving, and disseminating. Although associations among our predictors (risky sex, pornography use, and sexual preoccupation) and sexting have previously been investigated in isolation, it is not known to what extent associations among these variables overlap and to what degree sexual preoccupation uniquely predicts sexting behaviours, over and above other offline and online sexual behaviours.

We found significant positive associations among sexual preoccupation, pornography use, and risky sexual behaviours, supporting our Hypothesis 1. While bivariate correlations were low, these findings are consistent with previous literature, which noted associations among online sexual activities (e.g., pornography viewing), sexual preoccupation, and risky sexual behaviour [3,14,16,17,26–28,38,43,44]. Our results suggest that individuals who display a greater propensity for sexual fantasising, or preoccupation, are more likely to consume pornography and engage in risky sexual behaviours. As these associations were measured cross-sectionally, it is also possible that some people may be exposed to online sexual material first, eliciting excessive sexual thoughts that consequently lead to greater sexual exploration and offline sexual risk-taking. Regardless of the direction of these associations, the current findings indicate that, in the present day, online and offline sexual experiences are intertwined, and hence, both should be incorporated into public sexual and relationship education targeted towards young adults.

Hypotheses 2 and 3 were also supported, with sexual preoccupation significantly predicting both sending and receiving sexts, over and above other behaviours. Regarding the sending of sexts, our results are consistent with Howard et al.'s [32] finding that sexual preoccupation is uniquely associated with sending sexts. However, in our regression model, sexual preoccupation uniquely explained only 3% of the variance in sending sexts, whilst risky sexual behaviours and pornography use explained 19% of the variance. Specifically, pornography use and gender (women more likely to send sexts) were the strongest individual predictors of sending sexts. The fact that women were more likely to send sexts in our sample suggests that women may perceive sexting as a convenient platform through which they can flirt, express one's sexual fantasies, or arrange offline sexual encounters, with some degree of physical safety [56]. Further, given positive associations between pornography use and sexting, it is also possible that sexting constitutes a medium through which young people may replicate scripts regarding sexual behaviours and self-representation acquired through pornography viewing. Again, these findings illustrate that pornography use and risky sexual behaviours constitute the best markers of whether or not a person will engage in sending sexts. Therefore, sexual health education needs to incorporate the use of pornography and sexting, alongside the promotion of safer offline sexual practices, in order to encourage safe sexual behaviours in both contexts.

For receiving sexts (Hypothesis 3), again, sexual preoccupation was associated with an increased likelihood of receipt, but explained only 2% of the variance in our model. Pornography use and risky sexual behaviours explained 16% of the variance and were much stronger predictors, with pornography use being associated with a two-fold increase in both sending and receiving sexts. These findings are consistent with prior literature, where engagement in sexting has been associated with offline sexual behaviours [13–15], risky sex [16,17], and pornography consumption [29]. The fact that sexual preoccupation uniquely contributes to sending and receiving sexts suggests that for individuals preoccupied with sexual thoughts, sexting may constitute an important component of their sexual repertoire. Nonetheless, with pornography use and risky sexual behaviours being the strongest predictors of receiving sexts, public sexual education programs for adolescents and young adults need to transcend the promotion of safe offline sexual behaviours to incorporate online sexual behaviours, especially pornography use and sexting, highlighting their potential advantages as well as risks.

Our third analysis identified that sexual preoccupation was not uniquely associated with sext dissemination, thus failing to support our last prediction (Hypothesis 4). Our total model explained only 8% of the variance in dissemination, with gender (men more likely to disseminate) and risky sexual behaviour associated with dissemination. These results are consistent with Trendell [47] who also found no relationship between sexual compulsivity and dissemination. However, this contradicts previous research which suggested that sexual preoccupation is associated with sexual offending and recidivism [39–41,45,46]. Our results indicate that whilst sending and receiving sexts may be seen as extensions of regular online and offline sexual behaviours, with similar relationships with sexual preoccupation, dissemination to third parties may be motivated by less explicitly sexual and harmful reasons, with other motivations such as social status and humour noted in prior research [57].

### 4.1. Implications

Our findings contribute to a further understanding of the associations among potentially harmful online and offline sexual behaviours, particularly sexting exchanges, pornography use, and risky sexual behaviours, and the associations with sexual preoccupation. We found that sexting exchanges were associated with similar predictors and characteristics to other sexual behaviours (e.g., pornography viewing, sexual preoccupation). This suggests that many of the characteristics applicable to online sexual activities may also be relevant to sending and receiving sexts. More importantly, the fact that engagement in risky sexual behaviours was one of the strongest predictors of sending and receiving sexts increases the argument for education that explores the potential harms of risky online and offline sexual behaviours, including various forms of sexting. As sexting is becoming more frequent and ubiquitous, it is important that young people learn to use technology for sexual purposes in a responsible manner and that they potentially utilise it for the purposes of negotiating safer online and offline sexual practices.

### 4.2. Study Limitations and Directions for Future Research

Whilst our study presents novel findings, important limitations are noted. Firstly, the current study utilised a cross-sectional design; therefore, causal inferences among our variables cannot be made. While involvement in risky sexual behaviours, sexual preoccupation, and pornography viewing may predispose individuals to send sexts, it is also possible that this association is bidirectional, in that young people who are exposed to sexting may become more willing to explore and engage in various online and offline sexual activities. Future studies could utilise a longitudinal design whereby sexting, sexual risk-taking, and sexual preoccupation, for instance, are measured from early adolescence, allowing establishing the temporal and directional association among these variables. Secondly, our examination of sexting behaviours used dichotomous items assessing lifetime engagement in behaviours, consistent with the prior exploratory research methodology

used in this field [34,58,59]. However, our standardised measures of sexual preoccupation and pornography use frequency are expressed in the current tense, and the investigation of risky sexual behaviour focuses on the past year. As such, there is some variation in the timeframes of investigation, which may further impact the temporal relationships among variables. Future studies should ensure that all measures are temporally aligned. Further, our sample consisted of predominantly female Australian participants and is not population representative, as we drew on snowball recruitment. Therefore, the study findings cannot be generalised to the entire population. Future studies should investigate these variables in samples with a more equal gender distribution, as well as include more cultural diversity. Lastly, whilst we made all efforts in data collection to clarify that participation was anonymous, the potential social stigma associated with some behaviours may have influenced participants to underreport their sexual preoccupation or downplay rates of sext dissemination. In addition, to minimise survey burden, we did not include any instructional manipulation or attentional checks to verify responses.

## 5. Conclusions

This study found that sexual preoccupation was significantly associated with increased rates of pornography use, risky offline sexual behaviours, and both sending and receiving of sexts, whilst there was no association between sexual preoccupation and sext dissemination. In contrast to pornography viewing, which explained the most variance across all three sexting behaviours, sexual preoccupation made small unique contributions to our models. Nonetheless, our study illustrates that factors underlying online sexual activities are also applicable to sexting and that pornography viewing is the strongest predictor for engagement in sexting, for those examined in this study. Given positive associations among pornography use, risky sexual behaviours, and sexting, it is important that young people are educated on the harms associated with risky sexual behaviours in both online and offline contexts as part of respectful relationships and sexuality education programs. As the use and capabilities of modern technology are constantly evolving, it would be useful to teach young people how to take advantage of digital media to negotiate safer sexual practices, where issues such as consent could potentially be negotiated in an environment free from physical coercion or pressure.

**Author Contributions:** Conceptualization, E.M.C., D.H. and B.K.; methodology, E.M.C. and B.K.; formal analysis, E.M.C. and D.H.; investigation, E.M.C., D.H. and B.K.; data curation, E.M.C. and B.K.; writing—original draft preparation, E.M.C., D.H. and S.C.; writing—review and editing, E.M.C., D.H., S.C. and B.K.; supervision, B.K.; project administration, B.K. All authors have read and agreed to the published version of the manuscript.

**Funding:** This research received no external funding.

**Institutional Review Board Statement:** This study received approval from the Human Research Ethics Committee of Deakin University, under Reference Number 96-2012.

**Informed Consent Statement:** Informed consent was obtained from all subjects involved in the study.

**Data Availability Statement:** The data presented in this study are available on request from the corresponding author. The data are not publicly available due to ethical consent limitations.

**Acknowledgments:** The authors acknowledge the contribution by the students involved in recruitment for the broader study, from which the data for this report were extracted.

**Conflicts of Interest:** The authors declare no conflict of interest.

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
