# Peer review of "Dream It, Do It? Associations between Pornography Use, Risky Sexual Behaviour, Sexual Preoccupation and Sexting Behaviours among Young Australian Adults"

_sexes, doi:10.3390/sexes2040034_

Round 1

Reviewer 1 Report

I consider the research topic very up to date and important in terms of knowledge and organization for preventive strategies for risky sexual behavior.

Author Response

Thank you for your response and endorsement of our study. We appreciate your feedback and time.

Reviewer 2 Report

Dream it, do it? Associations between pornography use, risky sexual behaviour, sexual preoccupation and sexting behaviours among young Australian adults

This is an interesting, well written and scientifically sound contribution to the journal. Nevertheless, I believe some important changes would improve the overall quality of the paper:

  • Title: “Dream it, do it”? How does this interrogation contribute to the objectivity of the article?
  • Title: “Pornography” vs “Sexual Explicit Online Media” use. I encourage authors to clarify, since data was collected online.
  • Introduction: More information on pornography use in Australia among youth is necessary.
  • Title isn’t congruent with objectives, please confirm/correct: “this current study seeks to explore the associations between sexual preoccupation and sexting, pornography use, and risky sexual behaviour”.
  • Results: I suggest authors to provide readers with a table comparing results by sexual orientations.
  • Discussion: “Our study illustrates the need to incorporate pornography viewing and sexting into the promotion of safe sexual behaviours in on- and offline contexts, and the potential to utilise modern technology to negotiate safer sex”. Authors need to explain how this can be done.

Best wishes.

Author Response

This is an interesting, well written and scientifically sound contribution to the journal. Nevertheless, I believe some important changes would improve the overall quality of the paper:

  • Title: “Dream it, do it”? How does this interrogation contribute to the objectivity of the article?

Our intention here was to reflect our hypotheses and findings that engagement in virtual and imagined activities, and sexual preoccupation (“Dream it”), could then influence sexting behaviours (“Do it”). We believe that the second part of the title then clarifies this in more detail.

  • Title: “Pornography” vs “Sexual Explicit Online Media” use. I encourage authors to clarify, since data was collected online.

Thank-you for this comment. We note that our questions regarding both pornography use and sexual behaviours were not limited to online behaviours, but could be in any setting. As such, we feel that the title does reflect this accurately. We have modified the wording in the Measures section (Lines 218-221, page 5) to enhance clarity.

  • Introduction: More information on pornography use in Australia among youth is necessary.

Thank you for this suggestion. We have expanded our introduction regarding this topic, noting broad trends including both youth and population-representative adult data, where available (Page 2, Lines 90-103).  

  • Title isn’t congruent with objectives, please confirm/correct: “this current study seeks to explore the associations between sexual preoccupation and sexting, pornography use, and risky sexual behaviour”.

Thank you for pointing this out: We have clarified our statement of aims (Line 192-194), as our primary interest here was in determining factors associated with sexting behaviours, allowing for established relationships between our independent variables of sexual preoccupation, pornography use, and risky sexual behaviour.

  • Results: I suggest authors to provide readers with a table comparing results by sexual orientations.

Thank you for this suggestion. We have added comparisons by sexual orientation (coded dichotomously) to Table 1, and included commentary on this in our analysis (Lines 320-341). We also added sexual orientation into our correlations and regression analyses for completeness, but note that sexual orientation was not a significant unique predictor for any of our main analyses.

  • Discussion: “Our study illustrates the need to incorporate pornography viewing and sexting into the promotion of safe sexual behaviours in on- and offline contexts, and the potential to utilise modern technology to negotiate safer sex”. Authors need to explain how this can be done.

Thank you for this suggestion. We note that this phrasing was used in the Abstract. In the related Discussion section (Lines 660-662), we have expanded on this suggestion to clarify one way in which digital technologies could be used to negotiate consent and thus increase sexual safety.

Reviewer 3 Report

In my opinion this is an interesting study that may add to the scientific literature on sexting behaviours. However, there are several issues on the manuscript that makes replicability difficult. My principal concern is related with how sexting is measure and the sampling procedure and how it is explained. Following you can find some advice that you can follow to improve your paper:

Introduction:

It would be good that authors further elaborate the relationship between sexting, pornography and risky sexual behaviours. Authors mentioned them but the introduction is more devoted to analysing the link between sexting and sexual preoccupation.

Hypothesis should briefly recover the ground in what they are based.

Method:

A big issue is the insufficient description of the sample. Author/s should include a discussion of the desired sample based on a power analysis, then the procedure used (i.e., who was contacted about participation), and finally the number of participants who were involved in the study. How the authors established the representativeness of the sample?

When the online form was open and closed? How much time did participants employ filling the form? How long did the data collection process take? Was used an instructional manipulation check to verify that the participants had read the survey instructions and answer appropriately the questions in the online survey? How long did the data collection process take overall? Was there any missing data and how was it handled?

Measures:

Instead of using a sexting scale, sexting was measured through dichotomous items. This could be problematic. Moreover, sexting was measured without considering a timeframe while the other measures considered a timeframe of one year. The inconsistence in the timeframe makes difficult to understand the temporal relationship between the study variables. Authors should consider this limitation and explain how this could affect their results.

In the scales used, was conducted CFA in the current sample?

I find no weaknesses in the statistical analyses conducted.

Author Response

In my opinion this is an interesting study that may add to the scientific literature on sexting behaviours. However, there are several issues on the manuscript that makes replicability difficult. My principal concern is related with how sexting is measure and the sampling procedure and how it is explained. Following you can find some advice that you can follow to improve your paper:

  • Introduction: It would be good that authors further elaborate the relationship between sexting, pornography and risky sexual behaviours. Authors mentioned them but the introduction is more devoted to analysing the link between sexting and sexual preoccupation.

Thank you for this suggestion. We have extended this section to provide some further detail regarding relationships between sexting, pornography and risky sexual behaviours (Lines 78-86).

  • Hypothesis should briefly recover the ground in what they are based.

Thank you for this suggestion. We have added information to the hypotheses to summarise the core basis for these predictions (Lines 194-201).

  • Method: A big issue is the insufficient description of the sample. Author/s should include a discussion of the desired sample based on a power analysis, then the procedure used (i.e., who was contacted about participation), and finally the number of participants who were involved in the study. How the authors established the representativeness of the sample?

Thank you for raising these concerns. We have provided further details of the survey recruitment procedures, and the exclusion of participants not meeting our inclusion criteria due to age (Lines 303-306). We note that this study is not population representative, and have included this in our limitations (Lines 632-634).

  • When the online form was open and closed? How much time did participants employ filling the form? How long did the data collection process take? Was used an instructional manipulation check to verify that the participants had read the survey instructions and answer appropriately the questions in the online survey? How long did the data collection process take overall? Was there any missing data and how was it handle

Thank you for your questions regarding the survey process. This survey did not include manipulation or attentional checks, as the data was collected some time ago (noted in Lines 640-641). Additionally, and unfortunately, we are unable to answer some of these questions, as this data was collected some time ago, and the meta-data was deleted and cannot be retrieved from the server. We have noted the two participant responses removed due to inclusion/exclusion criteria (Lines 303-306), and that data collection was over an 8-week period (Line 295).

  • Measures: Instead of using a sexting scale, sexting was measured through dichotomous items. This could be problematic. Moreover, sexting was measured without considering a timeframe while the other measures considered a timeframe of one year. The inconsistence in the timeframe makes difficult to understand the temporal relationship between the study variables. Authors should consider this limitation and explain how this could affect their results.

Thank you for your valid concerns regarding the measurement of behaviours. We have noted this as a limitation and agree that in future, studies would benefit from a more consistent timeframe in the assessment of both independent and dependent variables. In relation to sexting specifically, many studies do measure sexting dichotomously, depending on the nature of the study aims and the construct of interest. In this case, as relationships between sexual pre-occupation and sexting behaviours overall had not been investigated, we wanted to establish this, prior to considering the extent or degree of sexting.

  • In the scales used, was conducted CFA in the current sample?

Thank you for this question. As all Independent Variable measures were drawn from existing and standardised measures (e.g. the Sex-Risk subscale of the Adolescent Risk Inventory (Lescano et al, 2007), the Pornography Consumption Questionnaire [Hald et al, 2006) and the Sexual Preoccupation Subscale of the Sexuality Scale (Snell & Papini, 1989)., we did not conduct a CFA as part of this study.

Hald, G.M. Gender differences in pornography consumption among young heterosexual Danish adults. Arch Sex Behav 2006, 35, 577-585, doi:10.1007/s10508-006-9064-0.

Lescano, C.M.; Hadley, W.S.; Beausoleil, N.I.; Brown, L.K.; D’eramo, D.; Zimskind, A. A Brief Screening Measure of Adolescent Risk Behavior. Child Psychiatry and Human Development 2007, 37, 325-336, doi:10.1007/s10578-006-0037-2.

Snell, W.E.; Papini, D.R. The sexuality scale: An instrument to measure sexual‐esteem, sexual‐depression, and sexual‐preoccupation. The Journal of Sex Research 1989, 26, 256-263, doi:10.1080/00224498909551510.

  • I find no weaknesses in the statistical analyses conducted.

Thank you for this and we are glad that you are satisfied with the analyses conducted.

Round 2

Reviewer 3 Report

I congratulate authors for this new version of the manuscript. I am recomending to accept the manuscript. However, my major concerns about how data was gathered and how sexting was measure remain present as major limitation of the current study. I do understand that authors are not able to improve these two major issues but the editor should consider if these are major flaws to publish the manuscript.

Author Response

Thanks to Reviewer 3 for this feedback and we note their continued concerns. As you state, it is not possible at this stage to alter the data collection and measures. We have noted these limitations in our paper (refer page 9-10). Specifically, we have noted the limitations in regard to differences in timeframes, and agree that in future, studies would benefit from a more consistent timeframe in the assessment of both independent and dependent variables.

In relation to the concerns regarding the measures of sexting, we have collected data in a manner consistent with other studies in the field (e.g., Choi et al., 2016; Del Rey et al., 2019; Gewirtz-Meydan et al., 2018; Temple & Choi, 2014). Specifically, we used a direct question relating to primary involvement in sexting. This is a common and accepted way to measure sexting behaviours (see References below for a few examples).

In general, exploratory studies where relationships have not previously been established do frequently use dichotomous and single-item measures, depending on the nature of the study aims and the construct of interest. In this case, as relationships between sexual pre-occupation and sexting behaviours overall had not been investigated, we wanted to establish this, prior to considering the extent or degree of sexting.

References

Choi, H., Van Ouytsel, J., & Temple, J. R. (2016). Association between sexting and sexual coercion among female adolescents. Journal of Adolescence, 53, 164-168. https://doi.org/10.1016/j.adolescence.2016.10.005

Del Rey, R., Ojeda, M., Casas, J. A., Mora-Merchán, J. A., & Elipe, P. (2019). Sexting Among Adolescents: The Emotional Impact and Influence of the Need for Popularity. Frontiers in psychology, 10, 1828-1828. https://doi.org/10.3389/fpsyg.2019.01828

Gewirtz-Meydan, A., Mitchell, K. J., & Rothman, E. F. (2018). What do kids think about sexting? Computers in Human Behavior, 86, 256-265. https://doi.org/10.1016/j.chb.2018.04.007

Temple, J. R., & Choi, H. (2014). Longitudinal association between teen sexting and sexual behavior. Pediatrics, 134(5), e1287-e1292. https://doi.org/10.1542/peds.2014-1974
